# Interplay between Prostate Cancer and Adipose Microenvironment: A Complex and Flexible Scenario

**DOI:** 10.3390/ijms231810762

**Published:** 2022-09-15

**Authors:** Mathilde Cancel, William Pouillot, Karine Mahéo, Alix Fontaine, David Crottès, Gaëlle Fromont

**Affiliations:** 1Inserm UMR1069, “Nutrition, Croissance et Cancer”, Université François Rabelais, Faculté de Médecine, 10 bd Tonnellé, 37032 Tours, France; 2Department of Medical Oncology, CHU Tours, bd Tonnellé, 37032 Tours, France; 3Department of Pathology, CHU Tours, bd Tonnellé, 37032 Tours, France

**Keywords:** prostate cancer, adipose tissue, fatty acids, persistant organic pollutants, adipokines

## Abstract

Adipose tissue is part of the prostate cancer (PCa) microenvironment not only in the periprostatic area, but also in the most frequent metastatic sites, such as bone marrow and pelvic lymph nodes. The involvement of periprostatic adipose tissue (PPAT) in the aggressiveness of PCa is strongly suggested by numerous studies. Many molecules play a role in the reciprocal interaction between adipocytes and PCa cells, including adipokines, hormones, lipids, and also lipophilic pollutants stored in adipocytes. The crosstalk has consequences not only on cancer cell growth and metastatic potential, but also on adipocytes. Although most of the molecules released by PPAT are likely to promote tumor growth and the migration of cancer cells, others, such as the adipokine adiponectin and the n-6 or n-3 polyunsaturated fatty acids (PUFAs), have been shown to have anti-tumor properties. The effects of PPAT on PCa cells might therefore depend on the balance between the pro- and anti-tumor components of PPAT. In addition, genetic and environmental factors involved in the risk and/or aggressiveness of PCa, including obesity and diet, are able to modulate the interactions between PPAT and cancer cells and their consequences on the growth and the metastatic potential of PCa.

## 1. Introduction

Prostate Cancer (PCa) is the most frequent non-cutaneous cancer in men and a leading cause of cancer death in western countries [1]. The prostate is surrounded by a rim of adipose tissue (AT), the periprostatic adipose tissue (PPAT), whose infiltration by cancer cells is linked to a poorer prognosis. White adipose tissue (WAT) is both an energy storage place and a metabolically active endocrine organ that secretes a variety of biologically active mediators contributing to paracrine and autocrine signaling pathways [2]. The involvement of PPAT in PCa aggressiveness has been strongly suggested by several studies, via production of hormones, inflammatory mediators, or lipid transfer [3]. In addition to PPAT, PCa cells are likely to be in contact with AT in the two most frequent metastatic sites, pelvic lymph nodes and bone, organs physiologically prone to adipose involution in elderly patients. In this setting, adipocytes are part of the tumor microenvironment (TME), together with many types of non-cancerous cells, such as fibroblasts, immune, and vascular cells. The crosstalk between TME and cancer cells can be direct by cell-to-cell contact, or paracrine via secreted molecules or exosomes [4]. The communication between cancer cells and AT is a two-way process that involves several players and mechanisms and leads to consequences in term of tumor aggressiveness and progression. Moreover, both genetic and environmental factors, including obesity, are able to modulate the interactions between WAT and PCa cells, as well as their consequences. Currently, several studies, either clinical or from in vitro research, have documented the relationship between AT and PCa. In this review, we present an up-to-date overview of the communication processes between PCa and AT, together with the identification of factors that can modulate this crosstalk, and therefore could be targeted for therapeutic interventions.

## 2. Cell Players

### 2.1. PCa Cells

PCa cells do not obey the classical Warburg effect phenotype observed in other solid tumors, oriented toward glucose uptake and aerobic glycolysis. In the early phases of PCa progression, the metabolism of cancer cells relies mainly on fatty acid (FA) oxidation [5], which is necessary for energy production and membrane synthesis to ensure cancer cell survival and proliferation. PCa cells can uptake circulating lipids or lipids from the adipose microenvironment to promote PCa invasiveness through oxidative stress in a NADPH oxidase–dependent manner, activating the Hypoxia Inducible Factor 1α/Matrix metalloproteinase 14 (HIF1α/MMP14) pathway [6,7]. It is only in the late stages of PCa progression that cancer cells will begin to exhibit the Warburg effect and have a higher rate of glucose uptake [5].

### 2.2. Adipocytes and Cancer-Associated-Adipocytes (CAA)

Adipocytes are the most frequent cell type in AT. A mature adipocyte is a large cell containing one lipidic vacuole made mainly of triacylglycerides [8], which is able to store and deliver lipids such as free FA to other cells, through lipolysis [9]. The adipocyte is a metabolically active cell, that can secrete numerous adipose-derived factors called adipokines, such as steroid hormones, growth factors including pro-angiogenic factors, chemokines, and cytokines, that are mainly pro-inflammatory [8,10]. Adipocytes interacting with cancer cells are called “cancer-associated adipocytes” (CAA), which are smaller than “standard” adipocytes, with a decreased lipid content, due to a phenomenon called delipidation [6]. Adipokines secreted by CAAs have an increased pro-inflammatory phenotype when compared to standard adipocytes [8,11].

Some morphologic and metabolic differences have been observed, depending on the site of AT. A study comparing PPAT to subcutaneous AT demonstrated that adipocytes were smaller in PPAT and secreted fewer polyunsaturated fatty acids (PUFAs), n-3 as well as n-6 [12]. Bone is an adipocyte-rich organ and bone marrow adipocytes (BMAds), which represent about 10% of the total fat mass, expand with aging [9,13]. BMAds, like WAT adipocytes, can store and release FAs and adipokines [13]. However, BMAds are devoid of lipolytic activity and exhibit a cholesterol-orientated metabolism [14].

### 2.3. Other Cells of the TME

Soluble factors can also be secreted by other components of the adipose TME, such as immune cells, cancer-associated-fibroblasts (CAFs), and mesenchymal stem cells (MSCs). CAFs predominantly originate from the local fibroblasts surrounding the tumor [15]. However, some studies also suggested that bone marrow-derived MSCs or adipocytes can be recruited to the TME and converted into CAFs [16]. MSCs are multipotent cells that reside in various areas such as bone marrow or fat, capable to differentiate into adipocytes as well as osteocytes or chondrocytes [17]. Pre-adipocytes, or adipose-derived stem cells, are highly represented in the AT stromal vascular fraction. This population is more frequently present in PCa than in normal adjacent prostate tissue [18] or in benign prostatic hyperplasia [19]. An in vitro experiment has shown that pre-adipocytes were more attracted toward Pca cells than non-tumoral prostate cells. After 48 h co-culture with pre-adipocytes, PCa cells displayed increased invasive properties [18].

## 3. Molecular Players

### 3.1. Lipids

Cancer cells activate lipolysis in tumor-surrounding adipocytes, leading to the release of free FAs. The main free FAs released are palmitic, oleic, and linoleic acids. Once released, free FAs are transferred to cancer cells [9]. A study using labelled FAs has demonstrated lipid-specific translocation between adipocytes and PCa cells [20].

Uptake of lipids in cancer cells has been shown to occur through either non-selective (macropinocytosis) or selective (lipid transporters) mechanisms [21]. Lipid transporters include the fatty acid binding proteins (FABPs) family, and the FA translocase CD36, expressed on PCa cells [22]. Targeting CD36 has been shown to reduce FA uptake and to slow cancer progression [22]. FABPs are a family of proteins that acts as intracellular FA transporters and are involved in the uptake and intracellular storage of FA [23]. FABP4 was first described in adipocytes, but is also expressed in other cell types, including PCa cells [23,24]. FABP4 has been shown to be up-regulated in human bone PCa metastasis, expressed in both cancer cells and adipocytes [25]. FABP4 is involved in lipid transfer between adipocytes and tumor cells, and is also able in PCa cells to translocate to the nucleus to interact with peroxisome proliferative-activated receptor γ (PPARγ) and affects cell growth and differentiation [24,26]. FABP5 is also involved in the uptake and transport of long chain FA, and its expression is upregulated in PCa tissues compared to normal prostate [23,27]. It has been shown that treatment with an FABP5 inhibitor suppressed the proliferation, migration, invasion, and colony formation of PC3-derived cells in vitro, by blocking FA stimulation of PPARγ [28]. Moreover, in a PC3 xenograft in mice prostate, the treatment with a FABP5 inhibitor prevented the development of metastases and induced a reduction in the size of primary tumors [28]. Otherwise, the role of exosomes in the communication between adipocytes and PCa cells has also been evidenced [29].

As demonstrated in patient-derived explants as well as in several cell lines, PCa cells uptake FAs to incorporate them into intracellular triacylglycerols (TAGs) and store them in the cytoplasm in the form of lipid droplets (LDs) [30]. LDs are made of neutral lipids, mostly composed of triglycerides (TG) and cholesterol esters (CE). In PCa cells, it has been shown that the amount of LDs increased with disease aggressiveness and progression [31]. FAs can be released from LDs either by lipolysis, which involves cytosolic lipases, or by lipophagy, a selective form of autophagy releasing free FAs that can fuel ß-oxidation in the mitochondria to provide energy [32]. A recent study has suggested an active process of lipophagy in PCa cells, linked to disease aggressiveness and to the proximity of cancer cells with PPAT [33]. FAs differentially influence PCa cell growth and migration. Animal and in vitro experiments have suggested that n-6 PUFAs stimulate, whereas n-3 PUFAs inhibit PCa growth [34,35]. Others have described a pro-migratory effect of saturated FAs, together with an anti-migratory effect of the n-6 PUFA linoleic acid (LA) and the n-3 PUFA ecosapentaenoic acid (EPA) [36,37]. It has also been shown that FA composition of PPAT in PCa patients is associated with markers of disease aggressiveness, like Gleason score and pathological stage. The FA profile associated with PCa aggressiveness was characterized by low level of LA and EPA, along with high levels of saturated FA [38]. In addition, in vitro migration potential of PCa cell lines supplemented with FA extracts obtained from PPAT was inversely correlated with adipose tissue LA content [38]. The effects of FAs released by AT on PCa cells is therefore likely to depend on the balance between pro- and anti-tumoral FAs.

AT is the primary cholesterol storage organ. Besides de novo cholesterol synthesis, PCa cells can uptake cholesterol from adipocytes in its free form, and store it as CE into the cytoplasmic LDs [39]. It has been shown that CE accumulation in cancer cells increased in high-grade PCa, via loss of PTEN and activation of PI3K/AKT [31]. Moreover, CE represents the main initial substrate for steroid biosynthesis [40]. It has been shown that ACAT-1, an enzyme essential for cellular cholesterol storage, and SREBP-2, a transcription factor involved in cholesterol uptake, are associated in PCa cells with the process of epithelial to mesenchymal transition (EMT), which allows cancer cells to acquire migratory and invasive characteristics, properties essential for dissemination and distant metastasis [41].

### 3.2. Hormones

WAT is an endocrine organ with the capacity to synthetize and metabolize steroid hormones. In men, it contains both androgens and androgen precursors that may support PCa growth and metastasis. Moreover, PPAT also expresses aromatase, an enzyme which converts androgens to estrogens [42,43]. Several lines of evidence suggest that estrogens may enhance PCa development and aggressiveness [44]. Estrogen synthesis in WAT increases with visceral adiposity, leading to an increase in free-circulating estrogen levels. In turn, estradiol induces a feedback inhibition of the pituitary–hypothalamic axis that decreases free testosterone levels [45].

### 3.3. Chemokines, Cytokines, and Proteases

Adipokines are cytokines, hormone-like polypeptides, secreted by adipocytes. They can exert their biological actions on target cells via endocrine, paracrine, and autocrine pathways [46]. The main adipokines known to play a role in PCa are leptin, adiponectin, interleukin-6 (IL-6), heparin-binding epidermal growth factor-like growth factor (HB-EGF), and vascular endothelial growth factor (VEGF) [46]. Circulating leptin has been shown to be two-fold higher in PCa patients compared to healthy people [47], and leptin receptor was detected in both normal prostate epithelial cells and PCa cells [48]. Moreover, leptin receptor expression on PCa cells has been associated with an adverse prognosis [49]. Recently, an in vivo study with LNCaP xenografts in mice, demonstrated that administration of a leptin receptor antagonist decreased tumor growth and delayed progression to castration-resistant disease [50].

Adiponectin has been shown to have effects which are clearly antagonistic to those produced by leptin, and thus may possess anticancer properties [46]. The expression of adiponectin receptors is decreased in PCa tissues compared to normal prostate [51], and is also decreased in aggressive PCa compared to indolent disease [52]. It has been suggested that adiponectin may play an essential role in suppressing PCa cells growth through inhibition of VEGF-A-mediated cancer neovascularization [51]. In AT, IL-6 is produced mainly by macrophages and stromal cells, with only about 10% synthesized by adipocytes [46]. IL-6 expression in PPAT has been correlated with markers of PCa aggressiveness [11]. Angiogenesis is the process by which new blood vessels are formed from preexisting vasculature, and has been shown to be correlated with high PCa stage and the presence of metastases [53,54,55]. A high degree of microvessel density, indicative of active angiogenesis, and high expression of VEGF, were found concurrently in high-grade PCa [55,56,57,58]. HB-EGF is an inducer of VEGF expression, and stimulates endothelial cell migration [46]. HB-EGF stimulates the growth of cultured human PCa cell lines, and is also produced by the tumor cells, providing the potential for significant autocrine mitogenic activity [59].

It has been demonstrated that mature adipocytes secrete the chemokine CCL7, which interacts with the CCR3 receptor on PCa cells to promote cell migration [60]. The same study also showed that the expression of CCR3 on PCa cells is associated with markers of disease aggressiveness. Moreover, CCR3 was found to be up-regulated in bone metastases compared to primary tumors [61]. In vitro experiments have shown that conditioned medium (CM) from human non-tumoral bone marrow derived adipocytes induced PCa cells migration, via the CCR3/CCL7 axis [61]. In vitro, CM from the PCa cell line DU145, which secretes CXCL1 and CXCL8, chemoattracts adipose stromal cells by signaling through their receptors CXCR1 and CXCR2, and promotes PCa progression [62]. CXCL12 is secreted by adipose stromal cells, and seems to play a role in chemoresistance as well as in PCa cell migration (LNCaP) [63]. In addition, PPAT can secrete some matrix metalloproteinases (MMPs), such as MMP9, which is able to induce invasive capacities in LNCaP cell line [64].

Some links have been identified between adipokines and hormone metabolism, since it has been shown that leptin significantly increases the expression of ER-α and decreases that of ER-β in PC-3 cells, and significantly up-regulates the expression of aromatase [65].

### 3.4. Persistent Organic Pollutants (POPs)

Many pollutants accumulate in AT due to their lipophilic nature; it is notably the case for different POPs [66]. POPs, as defined by the Stockholm Convention, are chemicals that have certain toxic properties, are resistant to degradation, and accumulate in living organisms. POPs are classified as polychlorobiphenyls (PCB), dioxins, and pesticides [66]. POP concentration in AT is more representative of the cumulative internal exposure than measurement in blood [67]. Some of these POPs have endocrine disruptive properties and can interact with hormone-sensitive cancers such as PCa [66,68]. Estrogen-mimetic properties, agonists against estrogen receptor alpha (ER-α), and antagonists against estrogen receptor beta (ER-β) have been demonstrated for some of them [69,70]. Moreover, chlordecone, an organochlorine pesticide previously used in the French West Indies, has been shown to exhibit in vitro affinity for the androgen receptor (AR) [70]. Another organochlorine pesticide, DDT and its metabolite DDE, displays anti-androgenic effects probably mediated by competitive binding to AR and/or inhibition of AR-dependent gene expression. Both DDT and DDE have been shown to repress PSA levels in human PCa cell lines [71]. Some epidemiological studies have identified a link between POPs concentration in blood and PCa risk [72,73]. In WAT, POPs can promote adipogenesis via estrogen or glucocorticoid receptors [66].

Some links have been identified between POPs and lipid metabolism. For example, chlordecone has been shown to modify tissue distribution of cholesterol, by decreasing the amount of CEs and intra-cellular LDs, through pregnane X receptor (PXR) and ER-α activation [74].

## 4. Scenario of the Interplay

### 4.1. On PCa Cells

Several pro-tumoral effects have been described in PCa cells, induced by adipocytes produced molecules, including proliferation, anti-apoptotic effects, invasiveness properties, and migration (Figure 1) [33,60,75,76]. Most of these effects have been observed for pro-inflammatory cytokines, chemokines, and lipids [37,49,77]. Increased migration capacities induced by FAs and leptin have been linked to the EMT process [37,49]. These effects have been shown to be mediated through several signaling pathways. In PCa cells, free FAs increase the expression of the pro-oxidant enzyme NADPH oxidase 5 (NOX5), leading to elevated reactive oxygen species (ROS). ROS activate the HIF1α/MMP14 signaling pathway, inducing increased invasiveness [6]. The effect of leptin to promote EMT in PCa is mediated by the stimulation of STAT3 pathway [49]. The involvement of the PI3K/Akt signaling has been demonstrated through inactivation of FOXO1 [78]. A process of adipocyte-induced lipophagy in cancer cells is also likely, since the expression of lipophagy markers on PCa cells has been shown to be higher in extra-prostatic areas of the tumor (in close contact with PPAT) than in intra-prostatic areas. In the same study, co-culture of PCa cells with adipocytes from PPAT induced increased lipophagy [33].

It has been shown that a high-fat diet was able to induce both lipid accumulation in prostate tumors and the development of metastases in a Pten-null mouse model of PCa [79]. In another study, high saturated fat intake induced proliferation in a PCa murine model via enhanced MYC transcription [80]. Some adipocyte secreted molecules are able to induce neuroendocrine differentiation (NED) in PCa cells. It has been shown that IL-6 treatment could lead to NED via increased lipid accumulation, PPAR-γ, and adipocyte differentiation-related protein (ADRP), a major component of adiposome. In addition, ADRP protein has been detected in exosomes released from these cells and these exosomes were able to induce NED in a paracrine fashion [29].

Moreover, anti-tumoral effects have been described with some adipocyte derived molecules, such as adiponectin [46], and PUFAs (Figure 1). Both LA and EPA have been shown to inhibit TGF-β induced PCa cell migration, via reduced Ca^2+^ entry regulated by SK3 (a potassium channel regulating calcium entry) and Zeb1 (an EMT transcription factor) [37].

The effects of PPAT on PCa cells could be influenced by both the quantity of AT and its inflammatory state. Several studies have found an association between PPAT thickness determined by MRI, and markers of PCa aggressiveness such as Gleason score and pathological staging [81,82]. Interestingly, no study has found a correlation between PPAT thickness and measurements of whole body adiposity measures such as body mass index (BMI) or waist circumference [81,82,83], suggesting that PPAT volume is independent of total adiposity and does not increase in the same relation than WAT in obese patients. PPAT inflammation, defined as the presence of dead or dying adipocytes surrounded by macrophages, has been associated with higher BMI and higher Gleason score [84]. The effects of secretome derived PPAT on PCa cells is therefore likely to depend on the balance between pro- and anti-inflammatory adipokines.

The effect of AT on PCa cells could also depend on the type of cancer cells. In vitro co-culture of mature adipocytes from rats with human PCa cell induced a pro-proliferative effect on the androgen dependent cell line LNCaP, whereas it had no effect on either proliferation or the PI3-kinase pathway in the androgen-independent cell lines PC3 and DU145 [85]. These data suggest that the pro-tumoral effect of adipocytes is most significant in differentiated hormone dependent cancer cells, whereas poorly differentiated androgen-independent PCa cells are more autonomous and could be less dependent on the adipocyte microenvironment.

### 4.2. On Adipocytes

Stimulation of PPAT explants with CM from the PC3 cell line has been shown to induce the secretion of pro-tumoral adipokines (osteopontin, TNFα and IL-6) and to reduce the expression of the protective adipokine adiponectin [86]. Moreover, co-culture of mature adipocytes with PCa cells induces transformation into CAAs that exhibit dedifferentiation, delipidation, and decreased size (Figure 1). Such morphological modifications have also been observed in vivo in the invasion front of human PCa where tumor cells are in close contact with adipocytes [6]. In addition, some POPs stored in WAT are considered as potential obesogens that could affect AT functioning. Epidemiological studies demonstrated an association between exposure to several POPs and higher BMI. POPs effects on adipocytes have been shown to be mediated through either endocrine disruption via fixation on nuclear receptors, or pro-inflammatory activity [87].

## 5. Factors That Can Modulate the Scenario

### 5.1. Obesity

Obesity has been linked to PCa aggressiveness [3]. Adipocytes from obese men are hypertrophic, with remodeling of the extracellular matrix that induces fibrosis and inflammation in the adipose microenvironment, leading to hypoxia and adipocyte stress. These changes lead to increase FA release and modification in adipokine production that favor the production of pro-inflammatory cytokines over anti-inflammatory cytokines (Figure 2) [88]. Higher secretion of the chemokine CCL7 has also been described in adipocytes from obese men compared to lean patients [60]. Moreover, obesity is associated with increased estrogen synthesis in WAT (Figure 2) [45]. More systemic disorders associated with obesity, such as dyslipidemia and insulin resistance leading to increased circulating levels of insulin, are also likely to be involved in PCa aggressiveness [3].

### 5.2. Genetic Factors

There are considerable ethnic disparities in PCa risk, with a 60% higher incidence rate among African-American men as compared with European-American men [89]. While such variations could be explained in part by differences in access to healthcare and PCa screening, it is now recognized that genetic factors play a determining role. Thus, certain genetic polymorphisms have been associated with an increased individual risk of PCa, and since their frequency varies according to the ethno-geographic origin of populations, they could help to explain the large variations in incidence [90]. The biology of PCa cells has also been shown to be different according to the ethnic origin of the patients (Figure 2). The TMPRSS2/ERG gene fusion, the most common genetic alteration in PCa, is twice as common in Caucasians as in African-Americans [91], and it has been suggested that the metabolism of FAs could be different in TMPRSS2/ERG positive and negative tumors [92].

Moreover, it has been shown that the FA composition of PPAT differed according to the ethno-geographic origin. In African-Caribbean patients compared with Caucasians, PPAT contained less monounsaturated and saturated FA, less n-3 PUFAs, and twice more n-6 PUFAs, mainly LA, an essential n-6 FA [38]. Since FA composition of AT reflects the past dietary intake, particularly for the essential FA, diet could therefore play a key role in the observed differences of PPAT composition. Alternatively, ethno-geographic differences could also be due to genetic variation in FA metabolism, since different variants in the fatty acid desaturases (FADs) gene have been evidenced in African-American men compared with Caucasians [93].

In addition, cholesterol metabolism could vary in cancer cells according to the ethnic origin (Figure 2). It has been shown that the expression of ABCA1 (involved in cholesterol efflux) was decrease in PCa cells from African-Caribbean patients compared to Caucasian patients, and that of SREBP-2 (involved in cholesterol uptake) was increased. This imbalance in cholesterol homeostasis oriented toward cholesterol accumulation in cancer cells was associated with a more frequent state of EMT, which may promote more aggressive PCa behavior [41].

### 5.3. Environmental Factors

As previously mentioned, dietary intake is likely to play a key role in the observed differences of FA composition in PPAT according to the ethno-geographic origin of the patients (Figure 2) [38]. The description of dietary habits in the West Indies reported high consumption of vegetable oils, mainly sunflower oil that contains about 60% of LA [94].

Lipophilic POPs, and particularly organochlorine pesticides (OCPs), have been shown to accumulate in fat, and their measurement in AT represents cumulative internal exposure [95]. Since food is the main vector of exposure to OCPs, diet plays a major role in human contamination (Figure 2) [96].

Finally, recreational physical activity after PCa diagnosis has been associated with a lower risk of PCa death [97,98]. Physical activity may reduce PCa aggressiveness through decreased body fat, improved insulin sensitivity, or decreased oxidative stress and systemic inflammation [99].

## 6. Concluding Remarks

A growing body of evidence supports bilateral communication between PCa cells and AT, particularly PPAT. Crosstalk with the adipose microenvironment is extremely complex, with consequences for cancer cells depending on both the amount and composition of AT. Indeed, the effects of PPAT on PCa cells are likely to depend on the balance between the pro- and anti-tumor components of the PPAT secretome. However, several interactions and mechanisms remain to be better understood, in particular the interactions between the different components of the PPAT secretome, using appropriate experimental models. The effects of bone marrow AT, which exhibits a different metabolism, also need further investigation. The effects of PPAT on PCa cells could also depend on the lipid metabolism of cancer cells, which is influenced by genetic factors, in particular the ethnic origin of the patients. The amount and composition of PPAT is highly dependent on targetable environmental factors, such as obesity, dietary FA intake, and exposure (mainly through diet) to POPs.

## Figures and Tables

**Figure 1 ijms-23-10762-f001:**
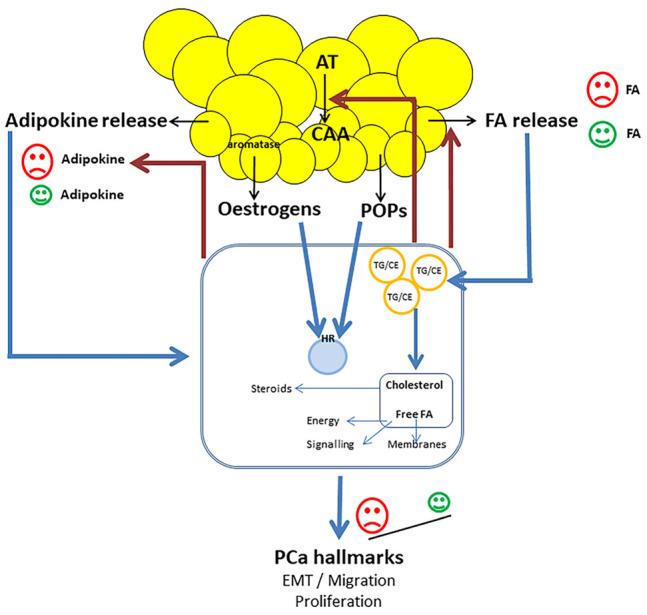
Crosstalk between PCa cells and periprostatic adipose tissue (PPAT): Adipocytes release not only adipokines and fatty acids (FA), but also estrogens and lipophilic persistent organic pollutants (POPs) stored in adipose tissue (AT). Both estrogens and POPs, which have for most of them endocrine disruptive properties, act on PCa cells through their effects on steroid nuclear receptors. FA are incorporated in cancer cells and stored in lipid droplets (LD), in the form of triglycerides (TG) associated with cholesterol ester (CE). Free FA can be released from LD, and then can either fuel β-oxidation in the mitochondria to provide energy, participate in cell signaling, or ensure membrane synthesis. In PCa cells, cholesterol is the main initial substrate for steroid synthesis. The effects of PPAT on cancer cells, and therefore on PCa hallmarks, probably depend on the balance between pro- and anti-tumor adipokines (for example leptin and adiponectin), and between pro- and anti-tumor FAs (for example saturated FAs and n-3 PUFAs). Cancer cells can induce both FA release from AT and the transformation of adipocytes into cancer-associated-adipocytes (CAA), that exhibit decreased size and delipidation. PCa cells can also induce the secretion of pro-tumor adipokines from adipocytes.

**Figure 2 ijms-23-10762-f002:**
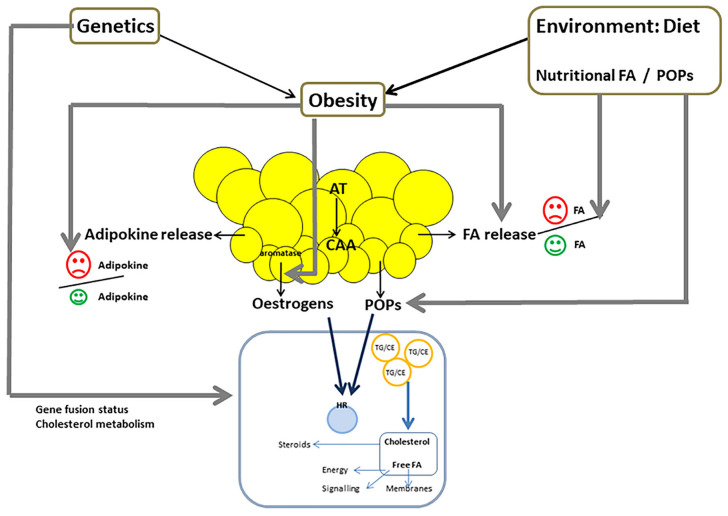
Factors that can impact the scenario of the interplay between periprostatic adipose tissue (PPAT) and PCa cells: Genetic factors, such as ethnicity, could modify the effect of PPAT on cancer cells, through different cholesterol and FA metabolism, mediated by the gene fusion status. Obesity induces increased FA release from adipocytes, increased production of pro-tumor adipokines, and decreased production of anti-tumor adipokines. Obesity is also associated with increased estrogen synthesis in adipose tissue. Diet, through FA and POPs intake, is also able to modify the effect of PPAT on cancer cells. Dietary intake plays a key role in FA composition of PPAT, and therefore impacts the balance between pro-tumor and anti-tumor FA.

## Data Availability

Not applicable.

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
