# Peer review of "Interplay between Prostate Cancer and Adipose Microenvironment: A Complex and Flexible Scenario"

_ijms, 2022, doi:10.3390/ijms231810762_

Round 1

Reviewer 1 Report

The authors have provided a comprehensive and interesting review on the interplay between prostate cancer cells and the adipose environment. The have discussed the composition of this interplay and various routes involved. This review could be a useful resource in the field to further understand the complexed prostate cancer tumor microenvironment. But it lacks insights of therapeutical options to exploit this interplay in order to achieve any clinical benefits

A few minor comments:

1, It should be noted that multiple studies have shown that lipogenic program drives prostate cancer progression. Some of the recent work such as  PMID: 29059155, PMID: 29335545, PMID: 30578319, PMID: 31554818 should be discussed. 

2, It would make the figures more completed if the names of anti- and pro-tumor FA and Adipokine are provided. And the figures should be referenced in the main text.

3, Typo in line 79

Author Response

Point 1: Two of those studies have been discussed in the revised manuscript, p10, l 4-7, and the references have been added.

Point 2: Some exemples of pro and anti tumor FAs and adipokines have been added in the Figure legend.

However, it is important to notice that, except for saturates and  n-3 PUFAs, the role of other specific FAs, particularly in the group of n-6 PUFAs remains to date controversed or unknown.

Point 3: Typo errors have been corrected.

Reviewer 2 Report

The present manuscript is a comprehensive review paper dealing with the interplay between prostate cancer (PCa) and adipose microenvironment (periprostatic adipose tissue, PPAT). The modulations of the interactions between PPAT and PCas by genetic and environmental factors (such as obesity and diet) were also discussed. There are some minor concerns as listed in the following:  

(1) It is better to have an Abbreviation List.

(2) Typos and others

*L4: Gaëlle Fromont1 and 3 .?

L124: ß-oxydation or ß-oxidation

L129: a pro-migratory effect of saturates?

L213: the french West Indies

L-357: ß-oxydation or ß-oxidation

*L516: R55: Vivo Athens Greece 2004, 18, 155–160 -> In Vivo 2004, 18, 155-160.

Author Response

Point 1: An abbreviation list has been added at the beginning of the revised manuscript.

Point2: typo errors mentionned by the reviewer have been corrected.